# Provisioning Australian Seed Carrot Agroecosystems with Non-Floral Habitat Provides Oviposition Sites for Crop-Pollinating Diptera

**DOI:** 10.3390/insects14050439

**Published:** 2023-05-04

**Authors:** Abby E. Davis, Lena Alice Schmidt, Samantha Harrington, Cameron Spurr, Romina Rader

**Affiliations:** 1Department of Environmental and Rural Science, University of New England, Armidale, NSW 2351, Australia; lschmidt@une.edu.au (L.A.S.); rrader@une.edu.au (R.R.); 2South Pacific Seeds, Griffith, NSW 2680, Australia; 3SeedPurity Pty Ltd., Margate, TAS 7054, Australia; cspurr@seedpurity.com

**Keywords:** non-bee pollinators, Syrphidae, pollinator management interventions, fly reproduction

## Abstract

**Simple Summary:**

Planting a diverse array of flowers in crop field settings can support insect crop pollinators, as many pollinating insects feed on floral pollen and nectar as adults. Although adult pollinating flies that feed on floral resources will also be supported by flower plantings, fly larvae rarely feed on floral nectar and pollen. Here, we deployed pools filled with habitats (decaying plant materials, soil, water) that pollinating fly larvae are known to feed on in seed carrot fields in an attempt to attract flies to lay eggs. We found many fly eggs and larvae within the pools after 12 to 21 days. More eggs were laid on decaying plant stems and carrot roots, compared to other locations (e.g., on decaying carrot flowers, leaves, etc.) within the pools. The habitat pools we trialed within the seed carrot fields could be a quick and easy way to support the reproduction of beneficial fly pollinators. These results can be used to support future studies to examine the effect of habitat pools in crop fields on the number of flies that visit crop flowers.

**Abstract:**

The addition of floral resources is a common intervention to support the adult life stages of key crop pollinators. Fly (Diptera) crop pollinators, however, typically do not require floral resources in their immature life stages and are likely not supported by this management intervention. Here, we deployed portable pools filled with habitat (decaying plant materials, soil, water) in seed carrot agroecosystems with the intention of providing reproduction sites for beneficial syrphid (tribe *Eristalini*) fly pollinators. Within 12 to 21 days after the pools were deployed, we found that the habitat pools supported the oviposition and larval development of two species of eristaline syrphid flies, *Eristalis tenax* (Linnaeus, 1758) and *Eristalinus punctulatus* (Macquart, 1847). Each habitat pool contained an average (±S.E.) of 547 ± 117 eristaline fly eggs and 50 ± 17 eristaline fly larvae. Additionally, we found significantly more eggs were laid on decaying plant stems and carrot roots compared to other locations within the pool habitat (e.g., on decaying carrot umbels, leaves, etc.). These results suggest that deploying habitat pools in agroecosystems can be a successful management intervention that rapidly facilitates fly pollinator reproduction. This method can be used to support future studies to determine if the addition of habitat resources on intensively cultivated farms increases flower visitation and crop pollination success by flies.

## 1. Introduction

The abundance and diversity of insects that provide pollination services within agroecosystems often depend on suitable habitat options for the insects to complete their life cycles [1,2]. These habitats (e.g., remnant vegetation, semi-natural landscape features) are typically not within the crop system itself, but nearby, and provide food, reproduction sites, overwintering resources, and shelter from agricultural management practices such as tilling, harvesting, and pesticide application [3,4,5,6,7]. When non-crop habitat is maintained or restored near intensely managed fields, beneficial insect abundance and diversity generally increase [4,8,9,10,11,12]. Even small patches (e.g., tens of square meters or less) of non-crop habitat can enhance beneficial insect biodiversity in cropping systems [3,4], and result in native species spillover into agroecosystems [4,13,14]. However, most pollinator-friendly habitat enhancements focus on floral interventions, such as floral strips and hedgerows [15,16,17,18], to attract adult, wild pollinators, primarily bees. Few studies have focused specifically on interventions to support the habitat needs for non-bee taxa (but see [2,19,20] for exceptions) and their non-floral resource needs (see [21] for an example of bee non-floral resource needs).

While bees are highly dependent on flowers to obtain nutrition for both adults and larvae, non-bee pollinator taxa, such as flies (Diptera), typically do not require floral resources in their larval stages [22]. For example, the larvae of eristaline syrphid flies, which are easily distinguishable due to the siphon-like ‘tail’ they use to breathe in poorly oxygenated habitats, live in wet substrates commonly found in agricultural environments, including decaying plant materials and manure [23,24,25]. Adult eristaline syrphid flies, like the cosmopolitan species *Eristalis tenax* (Linnaeus, 1758), have been shown to effectively pollinate crops as they morphologically resemble honeybees (*Apis mellifera* Linnaeus 1758) in size and body hairiness [26,27,28], despite lacking specialized pollen-collecting structures (e.g.,: scopa, corbicula). In fact, *E. tenax* is already a non-bee pollinator alternative in New Zealand, where the fly is an effective pollinator of multiple crops including seed carrot [29].

Seed carrot is an ideal model crop to study a potential non-bee pollinator since the crop often pollination limited despite attracting high numbers of other insect visitors [30,31]. Honeybees generally find seed carrot flowers unattractive, as the nectar composition is high in ferulic acid, an insect-feeding deterrent, and low in caffeic acid, a bee attractor [32]. As some species of eristaline flies have been shown to be as effective as honey bees at pollinating seed carrot [30,33], we hypothesized that building up populations of these beneficial non-bee pollinators could increase free ecosystem service delivery within seed carrot agroecosystems [29]. Therefore, this study was based in the Riverina region of New South Wales (NSW), Australia (AU), where seed carrot growers plan for the crop to bloom late austral spring and summer (November to December), when almost no other crops are concurrently blooming to best facilitate honeybee pollination services.

In this study, we trialed the deployment of small, portable pools filled with non-floral habitats to support the reproduction of eristaline syrphid flies in seed carrot agroecosystems. Although the life cycle of eristaline syrphid flies is generally well known [34,35,36], to our knowledge there are no studies that address whether eristaline flies have oviposition preferences within the habitat they utilize to lay eggs in natural field conditions. We, therefore, tested two habitats (soil with decaying carrot plants in water or decaying carrot plants in water only) to determine if existing adult eristaline syrphid flies would utilize the habitat pools as oviposition sites and evaluated where the flies oviposited within the habitat provided. We addressed the following research questions:
Will eristaline syrphid flies use provisioned habitat pools as oviposition sites in a commercial field setting otherwise unsuitable for larval development?Which of the two habitats resulted in the greatest number of eggs and larvae?What were the specific features within the habitat pools that resulted in the greatest oviposition?

## 2. Materials and Methods

### 2.1. Study Sites

Seven study fields, at a minimum of 315 m apart, of seed carrot monocultures in the Riverina region of NSW, AU, were chosen as sites in four locations comprising three commercial farms and one private farm managed by South Pacific Seeds (Figure 1). The seed carrot plots at the commercial farms varied from 5 to 14 hectares, while the private farm grew commercial-grade seed carrot in small (<500 m) trial rows. Other plant resources flowering nearby included onion (*Allium cepa* L.) at sites six and seven and small patches of native flowering trees, shrubs, and household gardens near sites one, four, and five as these sites were situated near residential areas. No other crops were observed flowering nearby. 

### 2.2. Deploying the Habitat Pools

In preliminary experiments at site one, we placed horse manure and wet, decaying carrot plants within a hybrid seed carrot plot to assess which substrates should be trialed as non-floral fly habitat. The substrates were observed frequently until we observed golden native droneflies, *Eristalinus punctulatus* (Macquart, 1847), oviposit within the wet, decaying carrot plants. As eristaline fly larvae are commonly reared in slurries of manure in laboratory conditions [24,36], this suggests that the larvae are more suited for semi-liquid environments. We did not trial manure at the field sites as golden native droneflies were not observed to oviposit within the manure and some landholders did not want manure placed on their properties; therefore, we chose to trial decaying carrot plants in water as reproduction sites for eristaline flies, with the presence or absence of farm soil. Thus, we hypothesized that more larvae would be found in the semi-liquid decaying carrot plant pool with soil, compared to the treatment with decaying plants and water only. 

Pools were deployed during peak bloom (50% flowering) of seed carrot (15 November to 9 December 2021), when adult eristaline flies are most likely to visit seed carrot flowers. Two polypropylene pools (945 mm × 210 mm × 1100 mm each) were placed side by side in a paired experimental design at each site to trial two habitats as eristaline fly oviposition sites (*n* = 7 per treatment, 14 pools in total). The first habitat consisted of soil, discarded carrot plants and water, while the second habitat consisted of discarded carrot plants and water only. Soil from the farm site was placed within the first pool until the base of the pool was covered, while the second pool contained two bricks to anchor the pool from strong gusts of wind that frequently blow within the region. Three fully grown (150 cm foliage height) male carrot plants were then taken from the site and placed in each pool which was then filled with the same water used to water the seed carrot crop (Figure 2a). We did not include a treatment without water, since eristaline flies cannot survive in habitats devoid of water [35,36]. Likewise, sufficient solid food must be present within the water for eristaline larvae to complete development [30], so we did not include a treatment of water-only pools. Instead, we conducted preliminary surveys searching for the immature stages of eristaline flies within field sites before pools were deployed, to confirm that no eggs and larvae had been laid in dry soil, dry plant material, or within crop rows. As no eggs or larvae were found in the preliminary field surveys, we excluded them from analyses.

After filling the pools with water, we left them undisturbed to allow the carrot plants within the pools to decay and for eristaline adults to locate the pools (Figure 2b). Due to unprecedented rain events at the time the pools were left undisturbed, site accessibility varied between farms; therefore, the pools were deployed for 12 to 21 days depending on field site accessibility.

### 2.3. Surveying the Immature Life Stages of Eristaline Flies

Once all farm sites were accessible on the same day, we conducted surveys in each pool to count eristaline syrphid fly egg clutches, a group of eggs laid together in a single oviposition attempt (Figure 3a), individual eggs, and larvae (Figure 3b). All egg clutches and individual eggs were counted on 9 December 2021 (starting at 06:00 and ending at 18:30) and were removed from the pools, so the eggs did not hatch before the larvae were counted. Additionally, we recorded the location of where the eggs were oviposited in the pools (Appendix A). All egg clutches and eggs that were displaced from their original positions in the pools (e.g., due to moving substrates) were counted but not included in statistical analyses. 

The day after the eggs were counted and collected, we returned to the pools and counted the eristaline fly larvae over a two-day period (with each pool counted only once for each immature life stage). To count the larvae, all plant material in the pools was thoroughly checked for individuals, and then removed from the pools. Next, we dislodged any larvae within the soil at the bottom of the soil treatment pools, by mixing the water with the soil sediment using a hand-held sieve. We then sifted the soil and water sediment through the sieve five times to determine the total number of larvae in the pools. When larvae were caught in the sieve, they were removed from the pools to avoid duplicate larval counts. For consistency, this procedure was also applied to the carrot and water treatment pools. We did not record where the larvae were found in the pools since we displaced all larvae when mixing the water with the sediment.

### 2.4. Rearing Eristaline Flies from Pools

Both eggs and larvae of various growth stages collected from each pool were reared to adulthood in controlled conditions on decaying carrot plants (from inside the pools) or a mixture of decaying carrot plants and sterilized horse manure to confirm species identities. Horse manure was mixed into the habitat as previous studies have successfully reared eristaline syrphid flies from manure [24,37]. Since eristaline syrphid flies have similar morphology in eggs and larvae and are therefore difficult to identify at these stages [23,37,38,39]; we waited until adults emerged to distinguish species identities using taxonomic keys [23,25].

### 2.5. Statistical Analyses

Statistical analyses were performed using R version 4.1.2. We created generalized linear mixed-effects models (GLMMs) using the *MASS* package to assess whether the number of eggs and larvae within pools differed based on treatment (two categories) or location (eight categories) [40,41]. To handle overdispersion in the collected count data, all GLMM models were fit to a negative binomial distribution [42]. Additionally, as the number of days the pools were left out in the sites to decay was not standardized as intended due to unprecedented weather conditions, we included the fixed effect ‘Day’ (continuous: 5 discrete values) in all models. We also included ‘Site’ as a random factor in all models, to account for the matched pair experimental design. 

The *DHARMa* package was employed on all models to perform residual, dispersion, and zero-inflation checks of the data [43]. For all significant models, we performed Tukey pairwise post hoc multiple comparisons tests between fixed effects using the *emmeans* package [44]. All figures were created using the *ggplot2* package [45].

## 3. Results

Two species of eristaline syrphid flies, the European dronefly, *E. tenax*, and the golden native dronefly, *E. punctulatus,* were reared out of both habitat pools at all seven sites. The fly *E. tenax* was successfully reared from all 14 pools, and *E. punctulatus* was reared from three of the 14 pools. Additionally, eggs and/or larvae of both species were found within all 14 pools (Table 1). The number of eggs within clutches ranged between 10 and 128 eggs (mean ± S.E. 54.7 ± 3.9 eggs/clutch) in the soil, decaying carrot plants, and water habitat and 15 to 125 eggs (mean ± S.E. 54.4 ± 3.6 eggs/clutch) in the decaying carrot plants and water only habitat.

Location within the habitat pools also influenced how many eggs were oviposited by female eristaline flies. We found significantly more eggs were oviposited within decaying carrot plant stems and decaying carrot vegetables compared to all other locations (Figure 4). There were no significant differences in the number of eggs laid within the pools based on habitat (*p* > 0.05 for both, Appendix A). Additionally, the number of days the pools were left out to decay did not significantly impact the number of eggs laid within habitat pools (z_1,4_ = −0.012, *p* = 0.91).

First, second, and third instar eristaline fly larvae were found within both habitat pools across all sites (Table 2). Significantly more living larvae were found compared to dead larvae (z_1,1_ = 6.13, *p* < 0.001); however, the longer the habitat pools were left out to decay, the fewer larvae of all three instars were found in the pools (Figure 5; see Appendix A for statistics). There were no significant differences in the number of larvae found within the pools based on habitat (z_1,1_ = −0.468, *p* = 0.64). Additionally, there were no larval instars more abundant than others within the habitat pools (*p* > 0.05 for all, Appendix A).

## 4. Discussion

In this study, we demonstrate that beneficial fly pollinators can utilize small, portable pools filled with locally available, cheap substrates (habitat) in seed carrot agroecosystems as oviposition sites. To our knowledge, no other study has deployed non-floral habitats to provide oviposition sites for pollinating eristaline flies. We found that all the habitat pools contained eggs or larvae of *E. tenax,* a cosmopolitan fly species that is an effective pollinator of carrot, onion, canola, and other cropping systems [28,46]. We also showed that the habitat pools encouraged oviposition by *E. punctulatus,* an eristaline fly endemic to the Australasian region [23]; however, other fly species in the genus *Eristalinus* Rondani, 1845 are known to be effective pollinators of other cropping systems including celery and fennel (Apiaceae), which are close relatives of seed carrot [47]. Species within the *Eristalinus* genus are found globally and have similar larval habitat and diet requirements [23,25,39]. Therefore, we predict that closely related flies from diverse biogeographical regions will be attracted to the habitat additions tested as well. Both fly species demonstrated oviposition preferences within the habitat, as more eggs were laid on decaying carrot stems, likely because this location within the pool offered protection for the eggs from the sun, preventing the eggs from desiccation, or from predators. 

We conducted this study to determine whether habitat pools could host eristaline fly reproduction. Thus, while it has been demonstrated that these pools act as oviposition sites for resident populations of flies, it is unclear how many pools are required to impact pollination services within different-sized fields. While we did observe *E. tenax* and *E. punctulatus* flies visiting seed carrot flowers in low numbers before these habitat pools were deployed within fields, it was beyond the scope of the study to compare the effect of habitat pool presence and absence on crop yield. 

It is well-known that many species of flies can develop in large numbers from small, transient resources [48,49,50]. The habitat pools tested were proven to be effective and were a quick and easy way to attract flies since both eggs and larvae were found within the habitat after a minimum of 12 days when the seed carrot crop was at peak bloom (50% flowering). For both fly species reared, the time needed to undergo different developmental stages is similar, as eggs hatch after 48 hours, and in optimal conditions, the larvae take an average of 12 days to develop before pupation [23,39]. Although the number of larval instars found did not differ significantly between pools, the majority had recently hatched and were in the first instar of development, and thus likely oviposited 48 to 96 hours previously. Hence, to best facilitate fly pollination during peak crop bloom, we suggest placing the pools nearby a different flowering crop or a small planting of flowers, 12 to 15 days before the flowering onset of the desired crop to ensure that adult eristaline flies locate the pools and two to three generations of syrphids emerge by the time the desired crop reaches peak bloom. 

Environmental conditions in the region at the time the habitat pools were deployed significantly influenced pool management. While these results suggest that the deployed habitat pools were low maintenance, we suspect that, under more average (i.e., drier, hotter) environmental conditions at this time of year within the region, this may not be the case. As the Riverina region is typically hot and dry in austral summer, we anticipated refilling the pools with water at least once or moving the pools to a shaded location to ensure the deployed habitat remained a suitable oviposition site for eristaline flies; however, rain events were common when we performed this experiment, so farms became inaccessible to check on the progress of the pools. As the mean temperature between November to December 2021 in the region was 21 °C to 24 °C, none of the habitat pools dried out completely; however, the pools were shallow and not completely shaded, so the sun could have heated up the habitat pools, which could have negatively affected egg and larvae survival. Therefore, we recommend deploying pools in completely shaded environments and monitoring the water level within pools, to ensure that the pools do not become ecological traps for eristaline flies [51].

Similarly, fewer larvae were found within the pools the longer the pools were left out to decay, suggesting the larvae within the pools left undisturbed for longer had either crawled out of the pools to pupate, died competing for food resources, or had been predated upon. As large amounts of decaying carrot plant debris were found within all pools, it is unlikely that the larvae died competing for food resources. The only observed predator within the pools was the rove beetle *Creophilus erythrocephalus* (Fabricius, 1775) which was present in two pools at the same farm. These rove beetles are known predators of fly larvae [52,53], although they have not been recorded feeding on rat-tailed maggots, specifically. Further research is required to better understand how to scale up these habitats to meet pollination service needs, the length of time the portable habitat pools should be placed on farms, the water conditions that eristaline syrphid fly larvae require to survive, the potential predators of the fly larvae, and whether these pools attract non-target or potential pest species to crop fields. 

## 5. Conclusions

In this study, we successfully trialed a non-floral resource habitat intervention which acted as oviposition sites for beneficial fly pollinators. These habitat pools are a unique, yet cheap, alternative for landholders who want to support pollinating fly reproduction but may not be able to set aside arable land for non-crop habitats. The flies oviposited within decaying carrot plant habitat, and larvae of all instars were found in pools within 12 days. Eristaline flies were found to preferentially oviposit underneath decaying plant stems, likely to protect eggs from predation or adverse environmental conditions. The substrates placed within the habitat pools (soil, discarded carrot plants, and water) are locally available, cheap, and the pools are small and portable, enabling placement and removal at key flowering times. This approach may increase the natural population of flies that provide critical pollination services to crops in intensely managed agricultural systems.

## Figures and Tables

**Figure 1 insects-14-00439-f001:**
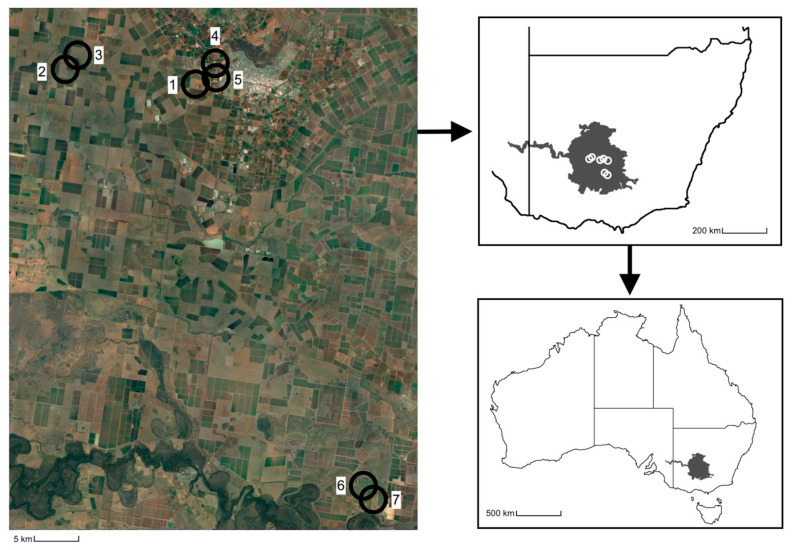
Location of the seed carrot study sites (1–7) where the habitat pools were deployed within the Riverina region of New South Wales, Australia.

**Figure 2 insects-14-00439-f002:**
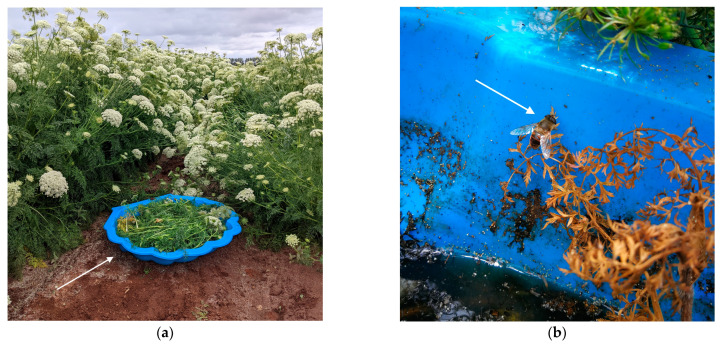
Experimental design of the habitat pools deployed to attract eristaline flies: (**a**) a habitat pool yet to be filled with water within a seed carrot field; (**b**) an adult, female *Eristalis tenax* (Linnaeus, 1758) fly within a deployed habitat pool. Arrowheads are pointing to the habitat pool and adult eristaline fly for clarity.

**Figure 3 insects-14-00439-f003:**
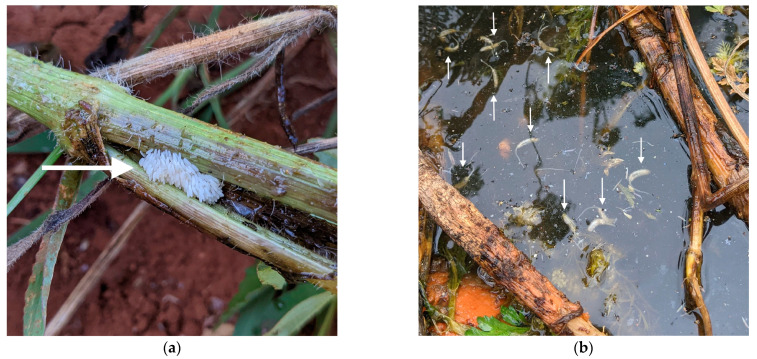
Immature stages of eristaline syrphid flies: (**a**) a clutch of eggs oviposited on a decaying carrot stem; (**b**) larvae found within a deployed habitat pool. Arrowheads are pointing to the immature stages of eristaline syrphid flies for clarity.

**Figure 4 insects-14-00439-f004:**
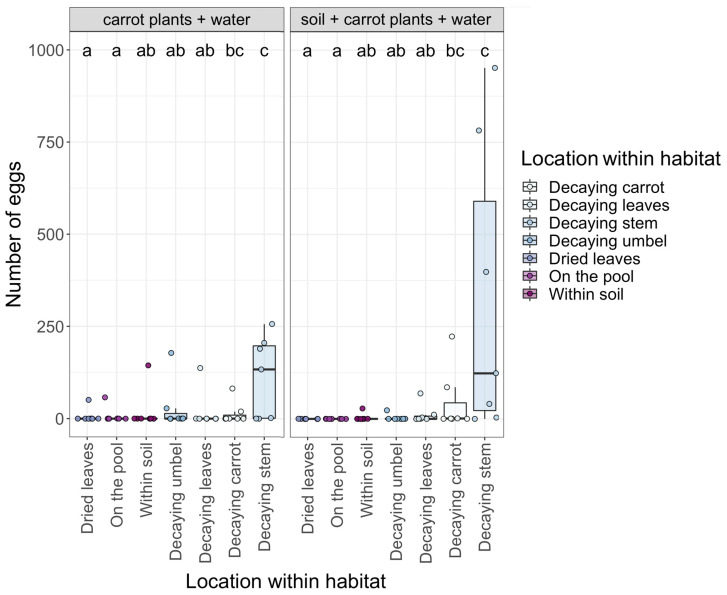
The number of eggs oviposited by female eristaline syrphid flies within the deployed pools based on habitat (carrot plants + water only and the soil + carrot plants + water) and the location where the eggs were laid. Letters indicate significant differences between locations (*p* < 0.05). Individual data points representing each habitat pool (*n* = 14 in total) are jittered onto the figure for clarity. Lower to upper box boundaries indicate the inter-quartile range (IQR). Whiskers are extended to the furthest data point within 1.5x the IQR from each box end.

**Figure 5 insects-14-00439-f005:**
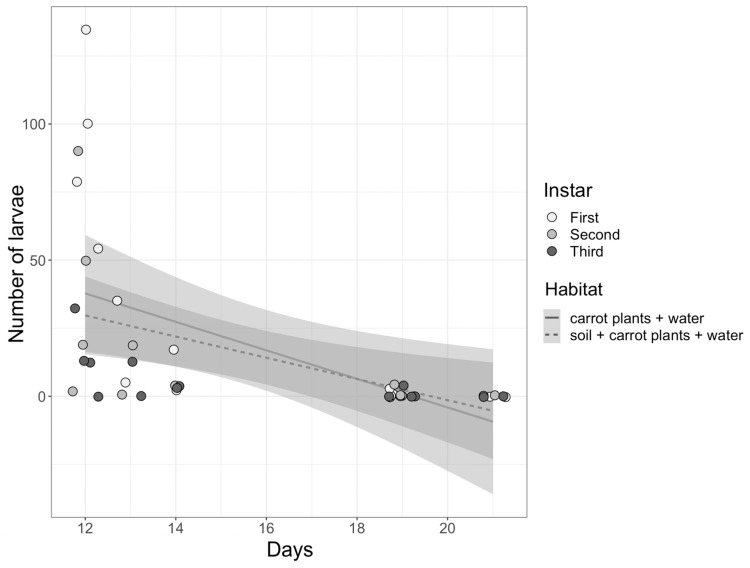
The number of larvae (first, second and third instar) found within the pools based on habitat (soil, carrot plants, and water or carrot plants and water only) and the number of days the habitat pools were left to decay. Individual data points representing larval instars found per pool (*n* = 14 per instar; 42 in total) are jittered onto the figure for clarity. The shaded outline of the linear regression lines indicates standard error.

**Table 1 insects-14-00439-t001:** Developmental stages of eristaline syrphid flies, *Eristalis tenax* (Linnaeus, 1758) and *Eristalinus punctulatus* (Macquart, 1847) found in two habitats (1 = soil, carrot plants, and water and 2 = carrot plants and water) at seven sites in the Riverina region of New South Wales, Australia. Both habitats were left to decay for a minimum of 12 days before surveying for fly egg clutches, eggs, and larvae.

Habitat	Site	Days	Clutches	Eggs	Larvae	Species
1	Site 1	21	16	910	0	*E. tenax*
1	Site 2	14	7	494	26	*E. tenax*
1	Site 3	13	6	296	41	*E. tenax*
1	Site 4	12	9	694	117	*E. tenax*, *E. punctulatus*
1	Site 5	12	22	1355	107	*E. tenax*
1	Site 6	19	9	382	9	*E. tenax*
1	Site 7	19	4	113	3	*E. tenax*
2	Site 1	21	9	476	0	*E. tenax*
2	Site 2	14	2	233	16	*E. tenax*
2	Site 3	13	0	0	41	*E. tenax*
2	Site 4	12	4	258	201	*E. tenax*, *E. punctulatus*
2	Site 5	12	21	1497	137	*E. tenax*, *E. punctulatus*
2	Site 6	19	6	548	4	*E. tenax*
2	Site 7	19	8	401	0	*E. tenax*

**Table 2 insects-14-00439-t002:** Total number of larvae found within habitat pools (1 = soil, carrot plants, and water and 2 = carrot plants and water) deployed at seven seed carrot sites in the Riverina region of New South Wales, Australia.

Habitat	Site	1st Instar	2nd Instar	3rd Instar	Dead
1	Site 1	0	0	0	0
1	Site 2	17	4	3	2
1	Site 3	35	1	0	5
1	Site 4	54	50	12	1
1	Site 5	100	0	0	7
1	Site 6	1	4	4	0
1	Site 7	3	0	0	0
2	Site 1	0	0	0	0
2	Site 2	2	4	4	6
2	Site 3	5	19	13	4
2	Site 4	79	90	32	0
2	Site 5	135	2	0	0
2	Site 6	0	0	0	4
2	Site 7	0	0	0	0

## Data Availability

The data presented in this study are available within the paper and its Appendix A.

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
