# Peer review of "Provisioning Australian Seed Carrot Agroecosystems with Non-Floral Habitat Provides Oviposition Sites for Crop-Pollinating Diptera"

_insects, 2023, doi:10.3390/insects14050439_

Round 1

Reviewer 1 Report

This study investigated the impact of pools to the reproduction of pollinating flies in seed carrot fields. It is a very original study, especially in the M&M deployement of the pools to test their impacts on non bee-insects

Do you have any idea of the survival from the eggs to the adult for the both species ? For your study, it is difficult to understand why do you choose to compare these two pool substrates (1= soil + carrot plants + water and 2 = carrot plants + water) ? In other words, what is the role of the soil in the deployed pools ? For Figure, you may use colors if you want.

It is well-written, and according to the originality of the study, I highly recommend this ms. 

Specific comments:

5: Prof. Romina Rader has none affiliation.

Abstract : 

14-15 + 26 : We do not understand the modality of pools 

L 30: Please add the year of the description of the scientific name, when first time in the text please check this in all the ms.

L28-31: I think that this sentence could be sliced in 2 or 3 smaller sentence, in a first read, this sentence is difficult to understand, as there are many combinations of dual objects "12 to 21 days", "oviposition and larval development",  "Eristalis tenax L. and Eristalinus punctulatus", "547 ± 117 fly eggs and 50 ± 17 larvae".

Introduction: 

I think that the second paragraph (L54 to 65) could go as first paragraph of the introduction.

L44-46 : I would not so sure about the lack of the reliability for the crop pollination service as wild bees, honey bees especially the generalist ones are always present in agricultural landscape.

I understand your point of view and agree with, so I suggest to replace "challenge" by "may challenge".

50: I think there are other ecosystem service such as water filtrering etc... please see this key reference to develop this https://www.sciencedirect.com/science/article/abs/pii/S0167880999000420

51-53: Can you explain it by an example ?

L111: In few words what are the two different substrate ?

M&M:

131-133: Can we have some pictures of the deployed pools on the field ? Could be add in the supplemental infortion

138-142: Ok, you cite the site number, but it is difficult to know where it is on the map of Figure 1.

148-150: Where ? on one of the seven selected sites

148-158 : I suggest that this part should go to the end of the introduction as it motivates the study.

162-164: Ok I understood now the both pools modality by reading Table 1 legend so this could be better explained in the text, pictures of the both modalities should be add in the ms as it is an original field experiment

182-190: How to be sure that you not missed some of the eristaline flower fly egg clutches as the eggs could have been hatched for 12 to 21 days ? Because you returned the day after to count the larvae ?

213: Can you specify how do you identify the adults ? Do you use taxonomic key, known insect collection, expertise of a taxonomist ?

216: You mean GLMM instead of GLMER (which I think this is the function of lme4 package in R ? https://www.rdocumentation.org/packages/lme4/versions/1.1-32/topics/glmer). As it is count data, why you do not transform your data by a log for example ? There are package that could give to you the best transformation for your data: bestNormalize package

Results:

Do you have the count of the adults surviving for both species ?

Discussion:

326-328: Another option is to build permanent "natural/artificial" ponds near seed carrot fields (or any other flowering crops which need the pollination of eristaline fly) and let for example some carrot plant decaying into it, what do you think about it ? Because I think that artificial pools cannot be applied at landscape scale of agricultural ecosystems as this could reinforce the consumption system in which we live.

Bibliography:

I think that some improvements need to be done here: italic for species names, one capital letter at the beginning of the citations ...

Reviewer 2 Report

Davis et al. report the results of a simple study on the effect of habitat provisioning on oviposition and larval development in syrphids in seed carrot fields. The study will be of interest to readers of Insects.

The study was carefully performed and is clearly reported and analyzed.

My only significant issue is the length of the Introduction and the Discussion. I think that the authors should focus on the main questions and main findings in both these sections. The text can be repetitive or excessively detailed. A significant reduction in the Introduction and Discussion would greatly improve the accessibility of the manuscript to its readers, in my opinion.

Of course, the English is fine; just a few grammatical errors and a few suggestions that I have written directly on a scanned copy of the ms.

Numbered points (see scanned ms)

1. The Introduction is far too long given the simple issue addressed in the study. Please reduce this by 50%.

2. This text should be moved to the Introduction.

3. Please give us an idea of how large the "fully grown" male plants were.

4. Filled with soil to a depth of....???

5. It sounds like you sampled ALL pools at all sites on the same day. Was this logistically possible? I would have thought that each pool took an hour or more to revise (at least!) x 7 sites and travel time over 100 km? This looks like a 2- or 3-day period of sampling.  Please clarify.

6. Suggest that you use arrowheads to indicate the eggs and larvae in Fig 2.

7. Ref 43 seems only to consider E. punctulatus. Were other keys used for species ID?

Line 230 correct typo in species name.

8. You use the term "substrate" when I would have used "habitat" at several points in the text. Sometimes you also use the term "treatment". As the treatments each involved two or three types of substrates, please consider how you used the term substrate to avoid confusion.

9. Better to say that the figs indicate "Numbers of eggs ....  (not totals)" and state what the bar, box and whisker indicate (median? IQR, range?)

10. I wondered if it would be better to place pools earlier alongside a small planting of flowering plants to allow time for 2 or 3 generations of syrphids before the carrot crop came into flower.  Is this unrealistic? Just an idea.

11. This is a long text on mosquitoes that do not seem to be an issue. It was unclear to me whether there are important public health mosquito-arbovirus risks in the study region.

12. This paragraph reads like a conclusion. Is it necessary given that it is followed by a conclusion section?

The references have some formatting issues. I think Insects uses abbreviated journal names (e.g. Proc. R. Soc. B). Volume numbers and article numbers or page numbers missing in some refs (although doi is given).

No worries.
